# Dielectrophoresis from the System’s Point of View: A Tale of Inhomogeneous Object Polarization, Mirror Charges, High Repelling and Snap-to-Surface Forces and Complex Trajectories Featuring Bifurcation Points and Watersheds

**DOI:** 10.3390/mi13071002

**Published:** 2022-06-26

**Authors:** Jan Gimsa, Michal M. Radai

**Affiliations:** 1Department of Biophysics, University of Rostock, Gertrudenstr. 11A, 18057 Rostock, Germany; 2Independent Researcher, HaPrachim 19, Ra’anana 4339963, Israel; michal.radai@gmail.com

**Keywords:** system’s perspective, MatLab^®^ model, microfluidics, DEP trajectory, LMEP, protein dielectrophoresis, virus trapping, LOC, μTAS, force spectroscopy

## Abstract

Microscopic objects change the apparent permittivity and conductivity of aqueous systems and thus their overall polarizability. In inhomogeneous fields, dielectrophoresis (DEP) increases the overall polarizability of the system by moving more highly polarizable objects or media to locations with a higher field. The DEP force is usually calculated from the object’s point of view using the interaction of the object’s induced dipole or multipole moments with the inducing field. Recently, we were able to derive the DEP force from the work required to charge suspension volumes with a single object moving in an inhomogeneous field. The capacitance of the volumes was described using Maxwell–Wagner’s mixing equation. Here, we generalize this system’s-point-of-view approach describing the overall polarizability of the whole DEP system as a function of the position of the object with a numerical “conductance field”. As an example, we consider high- and low conductive 200 µm 2D spheres in a square 1 × 1 mm chamber with plain-versus-pointed electrode configuration. For given starting points, the trajectories of the sphere and the corresponding DEP forces were calculated from the conductance gradients. The model describes watersheds; saddle points; attractive and repulsive forces in front of the pointed electrode, increased by factors >600 compared to forces in the chamber volume where the classical dipole approach remains applicable; and DEP motions with and against the field gradient under “positive DEP” conditions. We believe that our approach can explain experimental findings such as the accumulation of viruses and proteins, where the dipole approach cannot account for sufficiently high holding forces to defeat Brownian motion.

## 1. Introduction

Analytically, dielectrophoresis (DEP) is usually modeled using the electroquasistatic dipole approach [1]. There are few descriptions with the free energy approach or Maxwell’s stress tensor [2]. Almost every approach is from the object’s point of view. Recently, we presented a new analytical model from the system’s perspective. It is based on the capacitive charge work to suspend a single spherical object [3]. For DEP and the electro-orientation of ellipsoidal objects, our results have shown a steady increase in the overall polarizability of the suspension systems [4]. Even though this increase is slow with regard to the field oscillation, we propose considering electro-orientation and DEP as “conditioned polarization mechanisms.” Moreover, our results suggest that the law of maximum entropy production (LMEP) [5,6,7,8] provides a powerful phenomenological criterion for AC–electrokinetic effects. Our derivations have shown the importance of distinguishing between active and reactive components in DEP [3]. In the case of suspensions, the reactive components of the impedance result in extraordinarily high permittivities and conductivities at low and high frequencies, respectively. We suspect that this stealth effect has prevented—for over a century—any discussion of AC–electrokinetic forces in terms of the electrical work performed on suspensions.

In the dipole approach, the object is assumed to be small compared to the characteristic length of the field inhomogeneity. This allows the assumption of a homogeneous effective polarization described by the induced dipole moment with two equal dipole charges. Their interaction with the slightly inhomogeneous external field produces unequal forces at the two poles of the object, leading to DEP. The dependencies of the dipole moment on the field frequency and media parameters are summarized in the Clausius–Mossotti factor (CMF) [1,9]. Objects that are more and less polarizable than the suspension medium are assumed to move in (positive DEP) and against (negative DEP) the field gradient direction, respectively. This view is mainly correct for small objects. Objects of small size can “sense” the field gradient very locally and with negligible distortion of the external field. Consequently, their DEP trajectories “track” the steepest field gradient at each point.

However, in microchambers, complicated field distribution and inhomogeneous object polarization is typical, because the objects are relatively large with respect to the chamber [10,11,12,13,14,15]. The simple CMF description becomes problematic because the total force results from the superposition of polarization contributions from the entire volume of the inhomogeneously polarized object with the inhomogeneous field [16,17]. Examples include individual objects inducing mirror charges at the electrode surface or the attraction of two adjacent objects of the same size, where each object is subject to the field resulting from the inhomogeneous polarization of the respective other object. Analytically, these relationships are described by multipole models [18,19,20].

Our system’s approach shows how the DEP force can be derived from the charge work with an object moving between suspension volumes in an inhomogeneous field [3]. Maxwell–Wagner’s mixing equation described the apparent (or complex) specific conductivity of the volumes [21,22]. After separating the reactive and active components of the capacitive charge work, it could be shown that the active component drives the DEP [3]. At a given field frequency, the advancement of the object within the field gradient increases the overall polarizability of the DEP system through positive and negative DEPs in unison with its effective overall conductivity and, in turn, the dissipation of the electric field energy in ohmic heat.

In this paper, we generalize this approach by introducing a numerical “conductance matrix” to describe the overall (DC) polarizability of the system as a function of the position of the object in the DEP chamber. The assumption of DC properties for the object and the external medium prevents problems in separating reactive contributions in the electric work conducted on the DEP system [3]. However, it does not reduce the complexity of the field-induced object behavior since DEP is determined by the real (in-phase) part of the object polarization, even in the presence of complex media properties. 

As one example, we considered high- and low-conductive 200 µm 2D spheres in a square 1 × 1 mm DEP chamber with 199 by 199 “2D voxels” using the classical-plain versus pointed-electrode configuration. The conductance matrix contains the overall chamber conductances calculated for each position that was geometrically accessible to the sphere center. The matrix values were used as interpolation points for the MatLab^®^ quiver line function to calculate “conductance fields”, which completely describe the DEP behavior of the sphere. For a given start position, the complex trajectories of the sphere’s center follow the conductance gradient for the whole sphere, i.e., each step increases the overall conductance of the DEP system, and hence the dissipation of electric field energy at the fastest rate according to the LMEP.

## 2. Theory

### 2.1. General Remarks

The specific apparent, i.e., complex conductivity of aqueous media, is reduced by objects made of material with low conductivity or permittivity and increased in the presence of objects with high conductivity or permittivity. The actual effect is frequency-dependent. While the effective conductivity of suspensions increases with frequency, their effective permittivity drops [23]. Analytically, the conductivity of a suspension of monodisperse objects can be described by mixing equations [21]. For a given volume fraction, the effect of the objects on the conductivity of the suspension depends on their shape, orientation, and arrangement in relation to the external electric field [24,25,26,27,28]. 

The shape and frequency dependence of the induced dipole moment for objects confined by closed surfaces of the second degree (ellipsoids, spheroids, spheres, and cylinders) is generally summarized by the unitless, complex CMF, which has real (in-phase) and imaginary (out-of-phase) parts. For a homogeneous, general ellipsoid, it is described by the complex conductivities of the external $\left({\underline{\sigma }}_{e}\right)$ and object $\left({\underline{\sigma }}_{i}\right)$ media [29]: (1)$${\underline{f}}_{CM}=f_{CM}^{ℜ}+jf_{CM}^{ℑ}=\frac{{\underline{\sigma }}_{i}-{\underline{\sigma }}_{e}}{{\underline{\sigma }}_{e}+n\left({\underline{\sigma }}_{i}-{\underline{\sigma }}_{e}\right)}$$

Complex parameters are underscored. j being $\sqrt{-1}$. The ellipsoid’s shape is coded in the depolarizing coefficient n along the axis oriented in the field direction. For 3D and 2D spheres, it is 1/3 and 1/2, respectively. Note that the circle representing the sphere in 2D has the polarizability of a cylinder oriented perpendicular to the field in 3D [30]. 

The CMF is generally derived for a homogeneous external field, which induces a dipole moment. The DEP force is proportional to the real part of the CMF $f_{CM}^{ℜ}=ℜ({\underline{f}}_{CM})$, and any real polarization ratio of an object and external medium occurring for frequency-dependent properties can be modeled by combining appropriate DC conductivities for the external and object media. The imaginary part $f_{CM}^{ℑ}=ℑ({\underline{f}}_{CM})$ vanishes for the low- (ω→0) and high- (ω→∞) frequency limits, and the CMF is described by the real parts of the media’s conductivities ($\sigma_{i}$, $\sigma_{e}$) or permittivities ($\varepsilon_{i}$,$\varepsilon_{e}$). With n=1/2 for the 2D sphere: (2)$$f_{CM}^{ℜ}=\underset{\omega \to 0}{{\underline{f}}_{CM}}=2\frac{\sigma_{i}-\sigma_{e}}{\sigma_{i}+\sigma_{e}}\;\mathrm{or}\;f_{CM}^{ℜ}=\underset{\omega \to \infty }{{\underline{f}}_{CM}}=2\frac{\varepsilon_{i}-\varepsilon_{e}}{\varepsilon_{i}+\varepsilon_{e}},\;\mathrm{respectively}.$$

The same CMFs are obtained at the frequency limits for the same conductivity and permittivity ratios. The factors sweep the range between −2.0 and 2.0 (−1.5 and 3.0 for the 3D sphere) with the limiting values reached for $\sigma_{i}<<\sigma_{e}$ or $\varepsilon_{i}<<\varepsilon_{e}$, and $\sigma_{i}>>\sigma_{e}$ or $\varepsilon_{i}>>\varepsilon_{e}$, respectively. 

The CMFs of Equation (2) are three times larger than the usual expressions because the depolarization coefficient of 1/3 of the 3D sphere has not been separated and truncated against the 1/3 in the volume term; a step that is historically justified but is a simplification only for 3D spheres [9,17]. This allowed us to retain the full volume term in Equation (17), which reflects the ponderomotive (bodily) nature of the DEP force.

In the model below, we use the low-frequency limit by combining a tenfold ratio of external conductivity and object conductivity (1.0 S/m with 0.1 S/m and vice versa) corresponding to sphere and external medium conductances of 1.0 S and 0.1 S in 2D. These parameters yield CMFs of −1.64 and 1.64 for 2D spheres.

### 2.2. Charge Work and Conductance Change

A chamber of cuboid shape with two plain-parallel rectangular y by z electrodes of distance x is to be filled with a medium of complex specific conductivity ${\underline{\sigma }}_{e}$. The complex conductance of the chamber is:(3)$${\underline{L}}_{e}={\underline{\sigma }}_{e}\frac{yz}{x}=\left(\sigma_{e}+j\omega \varepsilon_{0}\varepsilon_{e}\right)k$$

$\sigma_{e}$ and $\varepsilon_{e}$ are the real parts of the conductivity and permittivity. ω and $\varepsilon_{0}$ being the circular frequency and the permittivity of vacuum. The cell constant k is the generalized geometry factor relating the conductance for chambers of any geometry to the conductivity of the measured medium. With a single object suspended at location i, the suspension’s effective conductivity is $\sigma_{S(i)}$ [3]. The chamber conductance is:(4)$${\underline{L}}_{S(i)}={\underline{\sigma }}_{S(i)}k=\left(\sigma_{S(i)}+j\omega \varepsilon_{0}\varepsilon_{S(i)}\right)k$$

Neglecting the stray capacitance, the same cell constant relates the suspension chamber’s capacitance to the suspension’s permittivity. The chamber can be described as a lossy capacitor, using the relation between complex conductance and capacitance:(5)$${\underline{C}}_{S(i)}=-j\frac{{\underline{L}}_{S(i)}}{\omega }=-j\frac{{\underline{\sigma }}_{S(i)}}{\omega }k=\left(\varepsilon_{0}\varepsilon_{S(i)}-j\frac{\sigma_{S(i)}}{\omega }\right)k$$

The charge work conducted on the capacitor by the rms AC-voltage $V_{eff}\;$ applied to the chamber is: (6)$$W_{S(i)}^{C}=\frac{ℜ\left({\underline{C}}_{S(i)}\right)}{2}V_{eff}^{2}=\frac{C_{S(i)}}{2}V_{eff}^{2}=\frac{\varepsilon_{0}\varepsilon_{S(i)}k}{2}V_{eff}^{2}$$

The energy (heat) dissipation in the chamber is:(7)$${\underline{P}}_{S(i)}={\underline{L}}_{S(i)}V_{eff}^{2}={\underline{\sigma }}_{S(i)}kV_{eff}^{2}$$

If the field at the object’s location is inhomogeneous, the capacitive charge work can induce DEP. In principle, the DEP work conducted in moving the object to location i+1 can be obtained from:(8)$$\Delta W_{}^{C}=W_{S(i+1)}^{C}-W_{S(i)}^{C}=\frac{C_{S(i+1)}-C_{S(i)}}{2}V_{eff}^{2}=\varepsilon_{0}\frac{\varepsilon_{S(i+1)}-\varepsilon_{S(i)}}{2}kV_{eff}^{2}$$

However, the description of the suspension properties, e.g., by mixing equations, may introduce reactive components and requires the identification of the active component of the apparent charge work, which drives DEP [3]:(9)$$\Delta W_{DEP}^{C}=\frac{\varepsilon_{0}}{2}\left(\varepsilon_{{}_{S(i+1)}}^{active}-\varepsilon_{{}_{S(i)}}^{active}\right)kV_{eff}^{2}=\frac{\varepsilon_{0}\Delta \varepsilon_{DEP}}{2}kV_{eff}^{2}$$

A first, a necessary but not sufficient condition for identifying $\Delta \varepsilon_{DEP}$ is that $\varepsilon_{{}_{S(i)}}^{active}$ and $\varepsilon_{{}_{S(i+1)}}^{active}$ are strictly the real components of permittivity. There are three ways to ensure that only active components enter Equation (9): analysis of the expression to eliminate reactive components and using the high-frequency limit of permittivity expressions or the low-frequency limit of conductivity expressions. For the two limiting cases, the reactive components of the DEP force vanish, similar to the imaginary components for ω→0 and ω→∞ in Equations (4) and (5), respectively [3]. For an illustration of the correspondence of the real, i.e., active parts of the suspension’s limiting permittivity and conductivity cases with the induced DEP force, see Figure 4 in [3]. DEP movement changes the dissipation by:(10)$$\Delta \underline{P}=\left({\underline{L}}_{{}_{S(i+1)}}^{}-{\underline{L}}_{{}_{S(i)}}^{}\right)V_{eff}^{2}=\left({\underline{\sigma }}_{S(i+1)}-{\underline{\sigma }}_{S(i)}\right)kV_{eff}^{2}$$

By analogy with Equation (9), the active component of dissipation is generated in the immediate vicinity of the object (compare with “influential-radius”) in phase with the external field. In contrast, the reactive component is generated by out-of-phase field components, mainly in the volume of the suspension medium. For ω→0, the (out-of-phase) reactive components vanish, and the apparent and active components of dissipation become identical [3]:(11)$$\Delta P_{DEP}=\left(\sigma_{{}_{S(i+1)}}^{active}-\sigma_{{}_{S(i)}}^{active}\right)kV_{eff}^{2}=\left(L_{{}_{S(i+1)}}^{active}-L_{{}_{S(i)}}^{active}\right)V_{eff}^{2}=\Delta L_{DEP}V_{eff}^{2}=\Delta LV_{eff}^{2}$$

The total dissipation of the DEP system becomes proportional to its DC conductance, simplifying the analysis of the DEP behavior.

### 2.3. DEP Force

For the fastest increase in the overall polarizability of the system and its active components, the DEP step from location i to i+1 must be oriented in the direction of the maximum differential quotient of the capacitive charge work or, more generally, the direction of the charge work gradient [3]. Using Equation (9) and the step width $\Delta r=\left|{\vec{r}}_{i+1}-{\vec{r}}_{i}\right|=r_{i+1}-r_{i}$ calculated from the location vectors ${\vec{r}}_{i}$ and ${\vec{r}}_{i+1}$, we obtain the DEP force: (12)$${\vec{F}}_{DEP}^{C}=grad\left(W_{DEP}^{C}\right)\approx MAX\left(\frac{\Delta W_{DEP}^{C}}{\left|{\vec{r}}_{i+1}-{\vec{r}}_{i}\right|}\right)\frac{{\vec{r}}_{i+1}-{\vec{r}}_{i}}{\left|{\vec{r}}_{i+1}-{\vec{r}}_{i}\right|}=\frac{\varepsilon_{0}k}{2}MAX\left(\frac{\Delta \varepsilon_{DEP}}{\Delta r}\right)V_{eff}^{2}\frac{{\vec{r}}_{i+1}-{\vec{r}}_{i}}{\Delta r}$$

$\frac{{\vec{r}}_{i+1}-{\vec{r}}_{i}}{\Delta r}$ defines the unit vector pointing in the direction of DEP translation. The DEP-induced differences in the active components of the charge work are always positive. At low frequency or DC, the increase in polarizability is strictly proportional to an increase in the conductance of the system and consequently dissipation. Accordingly, the DEP trajectory of a single object can be calculated from the maxima of the differential quotients of the DC conductance. In analogy to Equation (12), from Equation (11), we obtain:(13)$${\vec{F}}_{DEP}\sim grad\left(L_{DEP}\right)V_{eff}^{2}\approx MAX\left(\frac{\Delta L_{DEP}}{\Delta r}\right)V_{eff}^{2}\frac{{\vec{r}}_{i+1}-{\vec{r}}_{i}}{\Delta r}$$

Note that this relation between the DEP force and system polarizability coincides with the DC limit of Maxwell–Wagner’s mixture equation [22] (cf. [3]). To compare forces between different chamber and electrode setups, Equation (13) was normalized to the square of the chamber voltage and the basic conductance $L_{Basic}$ of the chamber without an object. In 3D, we obtain the normalized force: (14)$${\vec{F}}_{DEP}\sim \frac{1}{L_{Basic}}MAX\left(\frac{\Delta L_{DEP}}{\Delta r_{i}}\right)\frac{{\vec{r}}_{i+1}-{\vec{r}}_{i}}{\Delta r}$$

In the 2D description, fields, currents, etc., have no z-component, as in the case of thin films of uniform thickness, e.g., a metal layer on glass. Here, we use a 2D model geometry with a thickness of z=1m perpendicular to the sheet plane, neglecting the z-components. Combining the 3D suspension conductivity with thickness yields the DC-sheet conductances $L_{S(i)}^{2D}=\sigma_{{}_{S(i)}}^{active}z$ and $L_{S(i+1)}^{2D}=\sigma_{{}_{S(i+1)}}^{active}z$. Equation (11) reads: (15)$$\Delta P_{DEP}^{2D}=\Delta \sigma_{DEP}zk^{2D}V_{eff}^{2}=\Delta L_{DEP}^{2D}V_{eff}^{2}$$
with $k^{2D}$ being the 2D-cell constant. Note that both conductance differences, $\Delta L_{DEP}$ and $\Delta L_{DEP}^{2D}$ have the unit Siemens. In 2D, Equation (14) reads: (16)$${\vec{F}}_{DEP}^{2D}\sim \frac{1}{L_{Basic}^{2D}}MAX\left(\frac{\Delta L_{DEP}^{2D}}{\Delta r_{i}}\right)\frac{{\vec{r}}_{i+1}-{\vec{r}}_{i}}{\Delta r}$$

$L_{Basic}^{2D}$ is the system’s sheet conductance without an object. Note that the right-hand side of Equation (16) has unit “m”. The straightforward approach using the system’s capacitive-charging work (Equation (12)), similar to the derivation in [3], provides the “Newton” for the DEP force. Probably, from the object’s point of view, the correct proportionality factor in a 3D model includes the magnetic field constant. In the system approach used here, a normalized force is obtained, which can be converted into an exact force for a given location (see below).

## 3. Materials and Methods

### 3.1. Software

A 2D numerical solver based on the finite-volume method was implemented in MatLab^®^ (version R2018b). It was developed to simulate the potential distributions, current paths, and total conductance for arbitrary geometries and conductivity distributions with current sources (electrodes) [31]. 

The total conductance data for the 2D system with 199 × 199 2D voxels were stored in a matrix and used as interpolation points for the MatLab^®^ quiver line function to calculate the conductance field.

SigmaPlot 11.0 (Systat Software GmbH, Erkrath, Germany) was used for postprocessing and plotting data in line graphs. Inkscape 1.1.2 (GNU General Public License, version 3) was used to create graphical images and overlays of graphs with matrix images.

The data points of the Figures are given in the Appendix A. 

### 3.2. Numerical 2D Model

Without an object, a square chamber of x=y=1m confined by plain-parallel electrodes with a depth of 1 m perpendicular to the sheet plane has a (sheet) conductance of 0.1 and 1 S for volume conductivities of 0.1 and 1 S/m, respectively. The same sheet conductance occurs for square cm- or µm-size chambers with a depth of 1 m. Since only conductance and no size-related, frequency-dependent polarizabilities are considered, the 2D model is independent of a specific dimension in the x-y plane. To recognize microfluidic geometries, we assume an area of 1 × 1-mm^2^ for the DEP chamber, which is formed by 199 × 199 square elements. Each of the elements was assigned a homogeneous area conductance. Due to the assumed thickness, we refer to these elements as “2D voxels” (“2D volume pixels”).

The electrodes are located outside the chamber volume. The pointed and plain electrodes were formed by a single and row of 199 highly conductive 500-S voxels. The sheet conductance of the chamber was calculated for all positions accessible to a single 200 µm 2D sphere with a diameter of 39 voxels (Figure 1). The odd number symmetry defines a single central voxel and allows precise localization with respect to the pointed one-voxel electrode. Using 95 (19 voxels) and 497 µm (99 voxels) spheres, we showed that the results in this range do not qualitatively depend on the size of the sphere.

## 4. Result and Discussion

### 4.1. DEP Chamber Characterization

The classical setup with one pointed and one plain electrode was chosen to demonstrate the capability of our system approach. Figure 2A shows the field distribution without the sphere. Benign DEP behavior was observed in the range marked by the double arrow, which mainly corresponds to the dipole model. Figure 2B,C show the potential, field strength, and field gradient along the symmetry line. In Figure 2, Figure 3 and Figure 4, current lines were used instead of field lines to more clearly show the polarization of the sphere.

Theoretically, all plots in Figure 2 and the cell constant of the chamber $k^{2D}$ calculated from Equation (2) are independent of the medium conductance. The basic conductance values $L_{Basic}^{2D}$ of the chamber without an object were calculated with media of 0.1 S and 1.0 S from voltage and current using a MatLab^®^ routine. The two obtained cell constants showed negligible numerical differences.

### 4.2. DEP System with a Homogeneous Sphere

Figure 3 and Figure 4 show the field distributions in the DEP system for the two complementary conductance ratios of sphere and suspension medium for different sphere positions. According to theory, more favorable positions result in a higher overall conductance of the system.

### 4.3. Calculation of Trajectories and Forces

For a single-object suspension, the electric work conducted in the inhomogeneous field of a linear DEP system leads to the system’s overall permittivity and conductivity increase [3]. For objects with effective conductivities that are higher (“positive DEP”) and lower (“negative DEP”) than those of the external medium, this is true, even though the total conductance of the systems without the object is always lower or higher, respectively, than with the object (Figure 3 and Figure 4). 

The DEP behavior of the sphere was modeled using the “conductance matrix”. The 160 × 160 matrix elements were calculated as the overall sheet conductances of the system, with the sphere’s center located at each of the 160 × 160 accessible voxel coordinates. The basic sheet conductance determines the upper and lower boundary of the overall conductance of the DEP systems with the low- and high-conductance sphere, respectively. As a reference, the mean chamber conductance ${\bar{L}}^{2D}$ was calculated from all values in the conductance matrix. It corresponds to the average start conductance obtained in a field-free, thermally relaxed DEP system for infinitely many starting positions of the sphere. 

It was insufficient to consider DEP steps of the sphere’s center voxel to one of the up to eight neighboring voxels in the rectangular and diagonal directions to construct DEP trajectories from a given start voxel. We found that this “eight-neighbors” approach did not, for example, prevent the incorrect crossing of a bent watershed, i.e., bifurcation-boundary lines separating the catchment areas of different endpoints. Thus, we applied the quiver line function of MatLab^®^ to generate a “conductance field” using the elements of the conductance matrix as interpolation points. The conductance field provided smooth and more precise trajectories, watersheds, saddle points (bifurcation points), and normalized DEP forces. The program shifted the object stepwise along a quiver line in the direction of the maximum overall conductance increase to construct a trajectory. Positions with object voxels located outside the chamber area were excluded, i.e., the sphere was deflected by the chamber walls moving along the interface until reaching a point of attraction (endpoint). Clearly, the trajectories do not influence the endpoint conductance, while the DEP work conducted depends on the trajectory. Along each trajectory, the normalized DEP force was calculated with Equation (16).

One may ask whether the pointed electrode is very sharp compared with the experimental situation. However, there are at least two arguments against this assumption. First, in 3D, the 2D-pointed electrode corresponds to a 1 m vertical blade, and second, a voxel ratio of electrode to object of 1:39 (Figure 1) is similar to the size ratio of 110 nm-thick glass-chip electrodes and 4 µm cells [32].

### 4.4. Trajectories and Forces

Figure 5 and Figure 6 show the results for the two complementary conductance combinations. The 19-voxels-wide, white frames in Figure 5A and Figure 6A are geometrically inaccessible to the center of the sphere. In both conductance scenarios, the system’s sheet conductance increases steadily along each trajectory toward a specific endpoint (Figure 5B and Figure 6B). In Figure 5B,C and Figure 6B,C, sheet conductance and normalized DEP force, respectively, are plotted over the same abscissas.

At endpoint *E*_1_, the sheet conductance reaches a value close to ${\bar{L}}^{2D}$ (Figure 5B, insert) but more than twice that value at *E*_2_ (Figure 5B). At the instable saddle point *E_3_*, the DEP force vanishes. The sphere should theoretically travel along the watershed toward *E_3_* for start points on the watershed. However, a stable trajectory along the watershed could not be established for numerical reasons. 

In Figure 5A (trajectories b and e), geometric restriction by the chamber wall or plain electrode (trajectory *a*) causes deflections in the sphere’s trajectories. These are visible in the plots of the sheet conductance (Figure 5B) and, in particular, the normalized force (Figure 5C). The deflections occur before the sphere travels a longer distance along the plain electrode (trajectory *a*) or the chamber wall (trajectory e), or after it hits the restriction near an endpoint (*E*_2_, trajectories *b* and *c*). In the latter case, the maximum force is observed when the sphere touches the restriction. The final “correction” steps toward the endpoint generate a lower force. A similar process, although with a long “correction distance,” can be seen in trajectory *a* (Figure 5B,C, insert: green dashed curve). In the case of a “direct hit”, i.e., when the sphere reaches the endpoint directly (trajectories *d* and *f*), the force curve ends in the peak value. The peak forces reach very high values at the pointed electrode (*E*_2_) and moderate values when the sphere reaches the plain electrode (*E*_1_). 

The final steps along the projected conductance gradient before the vertical edge of the 2D sphere is attached to the plain electrode or chamber wall cause a greater increase in conductance and produce a higher force than the deflected motion in the attached state. However, minor movement in parallel to the restriction in the vicinity of an endpoint does not significantly change the sphere’s center distance to the tip of the pointed or center of the plain electrode. Thus, the overall conductance of the system does not change dramatically, and the resulting forces are not exceptionally high when the center or one of the neighboring voxels of the sphere’s edge touches the pointed electrode (*E*_2_) or the center voxel of the plain electrode (*E*_1_). Accordingly, the peaks with finally decreasing force (trajectories *b*, *c*, and *e*) can be explained by minor corrections of the position near an endpoint (Figure 5C). We suppose that such a force reduction after the final peak is also observed in high-resolution 3D models. 

However, the force curve here is probably additionally modulated by the shape approximation of the 2D sphere with straight vertical edges (Figure 1). Another effect that may play a role in this behavior was observed in 3D COMSOL Multiphysics^®^ (www.comsol.com) simulations with a highly conductive sphere in a coaxial DEP chamber. In this system, we found the conductivity minimum not in the attached state of the sphere but at a very short distance from the center electrode (results not published). For the model geometry used here, a comparable distance would be in the voxel size order of magnitude, preventing further investigation in this work.

Trajectories *d* and *f* run along the symmetry line between the pointed and plain electrodes. At saddle point E_3_, they start in opposite directions to different endpoints, although the common view would predict an attraction by the pointed electrode (cf. Figure 2). The existence of the watershed separating two regions of attraction along the symmetry line contradicts the dipole view.

In Figure 6, the sheet conductance of the system reaches slightly higher peak values at the “hidden” endpoints E_4_ and E_5_ than at the endpoints at the plain electrode. We suppose that the sphere’s size determines the shape of the watersheds and whether low-conductance spheres can “hide” away from the electrodes. The peak forces calculated for the 0.1 S sphere are generally lower than for the 1.0 S sphere.

### 4.5. Mirror Charge Effects

In the dipole model view, the 1.0 S sphere moves along the field gradient (Figure 2C), from the plain electrode to the pointed electrode, from areas with low field to those with high field. In our model, the sphere is initially attracted to the plain electrode (Figure 5A, trajectory *f*) and to the pointed electrode only beyond a watershed (Figure 5A, trajectory *d*). We suppose that the attraction toward the plain electrode is caused by mirror charges that exceed the dipole effect in the weak gradient in front of the plain electrode. 

In the dipole model view, the 0.1 S sphere moves against the field gradient (Figure 2C), from the pointed to the plain electrode, from high field to low field areas (Figure 6A, trajectory *f*). Above the first 100 µm, Figure 6C shows a very high, steadily decreasing repulsive force. However, the force increases again approx. 200 µm away from the plain electrode, despite the decreasing field gradient, until the sphere attaches to the electrode (insert of Figure 6C, trajectory *f*). The increase in force is related to the increase in the chamber conductance due to an overall reduced screening of the plain electrode by the low-conductive sphere (Figure 4A). This view is consistent with the interaction with mirror charges induced at the plain electrode. 

While the mirror charge effect clearly dominates for the 1.0 S sphere at distances of about 150 µm from the plain electrode, the observability of the 0.1 S sphere is blurred by the synchronous action of two equidirectional forces. We suggest that the mirror charge’s contribution to the DEP force depends strongly on the sizes and curvatures of the object and electrode. The electrode areas must be large relative to the object size for a high mirror charge to be induced. This effect is negligible at the pointed electrode.

### 4.6. DEP Force Reversibility in the Dipole Range

For 2D spheres or infinitely long 3D cylinders whose axis of symmetry is perpendicular to the field, the exchange of the conductivities of the external medium and the object reverses the sign of the CMF at constant magnitude (Equation (2)).

Accordingly, the direction of the dipole force is inverted for any position in the DEP chamber. If dipole forces prevailed, every trajectory would have to be exactly reversed, and the quotient of the DEP force magnitudes would have to be (minus) one. However, Figure 7 already shows a more complex picture along the symmetry line of the DEP chamber.

The plot of the quotient has characteristic regions separated by a zero point representing the vanishing force for the 1.0 S sphere at the watershed. The positive branch results from the attraction of the 1.0 S and 0.1 S spheres to the plain electrode. While the force magnitudes remain low in the positive branch, the quotient of seven at the plain electrode indicates the high-conductance sphere’s more efficient induction of mirror charges. Moreover, at the pointed electrode, this sphere experiences a force magnitude of around twice as high as the low-conductance sphere. From zero to the right, the force ratio reaches the expected −1 plateau, indicating dipole-like behavior. The plateau ranges from approx. −70 to 270 µm, i.e., over about 42% of the electrode distance along the symmetry axis accessible to the sphere. 

We suggest that reversibility is a criterion for the applicability of the dipole approach in certain regions of the DEP chamber. Nonreversibility indicates the presence of higher-order moments, mirror charges, and so on. Interestingly, the total conductance of the DEP chamber corresponds to the average conductance, roughly in the middle of the dipole region.

### 4.7. Relating Normalized to Actual DEP Forces

In the dipole region, the DEP forces only reach moderate magnitudes compared to the forces in front of the pointed electrode, where they reach magnitudes up to thousands of times higher than in the dipole region. Figure 8 considers the “dipole range” where the forces of the classical dipole model can be quantified and directly compared with the normalized DEP forces from Equation (16).

Stokes friction limits DEP velocities to the linear range in aqueous media, i.e., the velocities are proportional to the driving forces. Accordingly, experimentally observed accelerated DEP motion near electrodes is caused by changes in the DEP force. In the dipole model, the DEP force is:(17)$${\vec{F}}_{DEP}=ℜ\left(\underline{\vec{m}}\right)\cdot grad\left(\vec{E}\right)=\varepsilon_{0}\varepsilon_{e}V_{0}f_{CM}^{ℜ}\vec{E}\cdot grad\left(\vec{E}\right)$$
where $\varepsilon_{0}$ and $V_{0}$ are the permittivity of vacuum and the volume of an ellipsoidal object:(18)$$V_{0}=\frac{4\pi }{3}abc$$
with the principal semiaxes a, b, and c (2D sphere: a=b, c=1 m). For an electrode voltage of 1 V and the field parameters of Figure 8, we obtain $\vec{E}\cdot grad\left(\vec{E}\right)=0.5602\;V^{2}/m^{3}$, which can easily be rescaled to any experimental electrode voltage. Experiments or calculations can provide values for the actual DEP force at the chamber position x = 186 µm, y = 0 µm, where our model yields normalized forces of approx. 0.12 for the 2D sphere. Clearly, the 2D inhomogeneity in the polarization of the 2D sphere causes the very high force magnitudes at the pointed electrode (Figure 3D, Figure 4D, Figure 5C and Figure 6C). They are about 1500 (1.0 S sphere) and 580 (0.1 S sphere) times higher than the forces in the dipole region at the chamber position considered.

The exact force conversion with Equation (17) indeed calls for calculating the conductance field for a 3D object. However, in dipole theory, the force magnitude experienced by a 3D sphere in positive DEP can be twice that experienced in negative DEPs (Equations (1) and (17)). Moreover, we believe that 3D inhomogeneity would induce even higher forces at the pointed electrode when polarizing a 3D sphere.

### 4.8. Remark on “Positive” and “Negative” DEP

Trajectories with a changing orientation within an inhomogeneous field indicate a problem with defining the sign of the DEP force in relation to the external field gradient. While “positive” and “negative” DEPs correspond to the common understanding in the dipole region, the definition becomes fuzzy elsewhere in the chamber, especially when the force direction reverses because the object itself changes its “field environment,”, e.g., due to mirror charges. 

### 4.9. Thermodynamic Aspects

The theoretical description of AC–electrokinetic effects, such as electro-orientation, DEP, electrorotation, or mutual attraction, usually relies on electrostatic approaches. However, for lossy media, the validity of the approach is not clear per se, since electrostatic systems are generally in a state of equilibrium without energy dissipation and entropy production by resistive and displacement currents. Moreover, the electrokinetic effects induced must themselves lead to energy dissipation. Despite these seemingly severe problems, our LMEP approach and experimental observations in the dipole region agree very closely with object-oriented electrostatic models.

Assuming that the DEP system is near equilibrium in its linear range and the application of the electrode voltage causes only minor deflection of the system from equilibrium, which remains in the linear range, it should approach a new “voltage-on equilibrium” through the minimization of entropy production according to Prigogine’s principle [33,34,35]. Then, after voltage-off, the system should return to its previous state. Nevertheless, our theoretical and experimental findings suggest that electrokinetic phenomena increase the overall energy dissipation and are in contradiction with Prigogine’s principle. In fact, the problem had already been addressed in 1912 and discussed by [36] (see also the references contained therein). In light of this work, one may wonder how a Kirchhoff network, which is generally the electrotechnical basis of our approach, would react if it could rearrange itself. Perhaps a kind of “Kirchhoff Rearrangement” is the effect we observe with the DEP?

From the point of view of the system, the work on a volume of material can be stored or dissipated (i) as electric field energy, (ii) as magnetic field energy, or (iii) as Joule heat [29]. In our model, we use the degree of the DEP system’s overall increase in conductance as a criterion for the DEP force induced. At constant electrode voltage, dissipation of electrical energy is proportional to the square of the applied voltage according to Rayleigh’s dissipation function, and it increases with the total conductance of the system [37]. While a small proportion of this energy is “dissipated” in DEP translation, DEP increases the total energy dissipation and the electrical work that must be done while DEP progresses.

In particular, it has been shown how DEP is related to the complex, i.e., apparent permittivity and conductivity of the suspension, both of which consist of an active and a reactive part. Like electrical machines, the reactive part (capacitively stored at the objects) is out of phase with the active component and performs no DEP work. While the DEP force is proportional to permittivity and conductivity’s active components, the reactive components are dissipated [3]. A related discussion on the contributions of electronic polarization to the total field energy in lossy dielectrics seems to be underway [38].

### 4.10. Nonspherical Objects

An applied field can change the impedance of a suspension by inducing electro-orientation, DEP translation, or electrodeformation of objects [11,39]. As shown in theory and experiments, the (frequency-dependent) axis with the highest CMF is aligned in linear fields [32,40]. In homogeneous ellipsoids, the longest axis always has the highest CMF and is aligned [27]. This leads to a reduced suspension impedance and, at constant field strength, increased electrical power dissipation. For homogeneous spheroids, it has been theoretically demonstrated that the field-induced orientation moments are proportional to the increase in the conductivity of the suspension they induce [4].

A nonspherical object in an inhomogeneous field experiences force and torque, simultaneously resulting in both DEP and electro-orientation. Furthermore, the object’s movement modifies both force and torque. Friction opposes both types of motion and can, for example, prevent a complete alignment at a particular location before the object moves to another location where a different alignment is induced. The situation is further complicated by the different nature of the friction opposing the translational and rotational motion, as can be seen, for example, from the different radius dependencies for the translation (~R: Stokes friction) and rotation of spheres (~R^3^) [41].

## 5. Conclusions and Outlook

Recently, we derived the classical DEP force expression from the capacitive charge work gradient on a suspension of a single object in an inhomogeneous field, but abstracting from the actual chamber and electrode geometries. Here, we extended this approach to the entire DEP chamber by introducing a conductance field, the low-frequency equivalent of the capacitance field. The fields fully describe the object’s DEP behavior and inherently account for inhomogeneous object polarization, mirror charges, electrode shielding effects, and so on. 

Our model simplifies the computation of DEP forces in complex field environments. However, if the approach is applied to nonspherical objects or multibody systems, for example, to compute aggregation patterns, this comes at the expense of high computational effort, especially in 3D systems. Appropriate methods, such as Monte Carlo simulations, would likely reduce the computation time. 

Objects with an effective conductivity lower or higher than that of the suspension medium usually show negative or positive DEPs, in other words, they move counter to or in the direction of the field gradient. Here, we reveal some exceptions to this rule outside regions where dipole effects dominate, something that may call for a conceptual rethink. A manuscript is in preparation extending the present results to pointed-versus-pointed, plain-versus-plain electrodes, and to four-pointed-electrode arrangements in one-versus-three, side-by-side, and field-cage drive modes.

We believe that our model can explain experimental findings such as the paradoxical accumulation of viruses and proteins in field cages or at electrode edges, where the dipole approach cannot account for sufficiently high trapping forces to withstand Brownian motion [16,17,42,43,44]. Forces large enough to trap small objects can result from inhomogeneous object polarization at electrodes or other surfaces and near to other objects. Although these forces act over larger distances than the Van der Waals forces do, the distances appear to be too small to trap objects from the entire suspension volume. This is why we propose the “sticky-fly-trap model”. Viruses or molecules that approach electrode surfaces or other viruses or molecules by media flow or diffusion, to the point of becoming inhomogeneously polarized, snap to the surface or can form aggregates. This mechanism is suggested by experiments, which show that the aggregation of objects takes longer than redispersion after the field has been turned off. The aggregation is limited by the “undirected”, random motion, while the dissolution of the aggregate is achieved by a “directed” diffusion away from the high concentration.

One final point is worth mentioning: the striking similarities between the snap-to-surface behavior in the DEP model and in our earlier force spectroscopy experiments with like-charged glass spheres and mica surfaces [45]. In these experiments, snap to surface has been observed over distances of up to 800 nm, clearly too large to be explained by the Van der Waals attraction, although this may lock the bead at the surface once it is reached. Here, we propose a “self-DEP” mechanism to explain the large snap-to-surface distances. Self-DEP can result from the oscillating dipole induced in a vibrating bead with fixed surface charges, e.g., in the mid-kHz range. In the case of dispersion, the vibration leads to the induction of a dipole by separating the center of the bead’s fixed charges, which move with the object, and the center of the countercharges in the external medium. In “self-DEP”, the interaction of the oscillating dipole with mirror charges induced on conducting or polarizable surfaces leads to attractive forces. The contribution of mirror charges to DEP forces has been described by Pethig; chapter 5.4 in [46]. Here, we found the attraction by mirror charges for both high- and low-polarizable objects.

The systems perspective allowed us to identify new approaches and perhaps even new fields of work for DEP research. These are (i) the modeling of the high repulsion and snap-to-surface forces induced by the inhomogeneous polarization of the objects; (ii) the calculation of DEP forces in complex field environments and multibody systems; and (iii) the role of active and reactive contributions in frequency-dependent DEP and other electrokinetic AC models in relation to the total work done on the systems [3].

## Figures and Tables

**Figure 1 micromachines-13-01002-f001:**
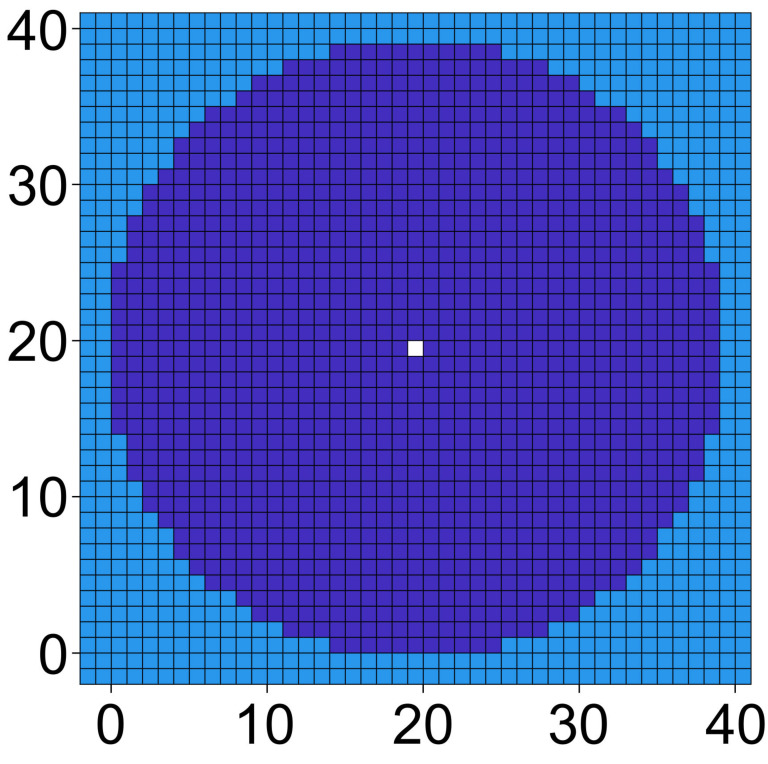
200 µm 2D sphere approximated with a diameter of 39 voxels in the horizontal and vertical directions. The central voxel is marked in white. Linear rows of 11 voxels form the horizontal and vertical edges.

**Figure 2 micromachines-13-01002-f002:**
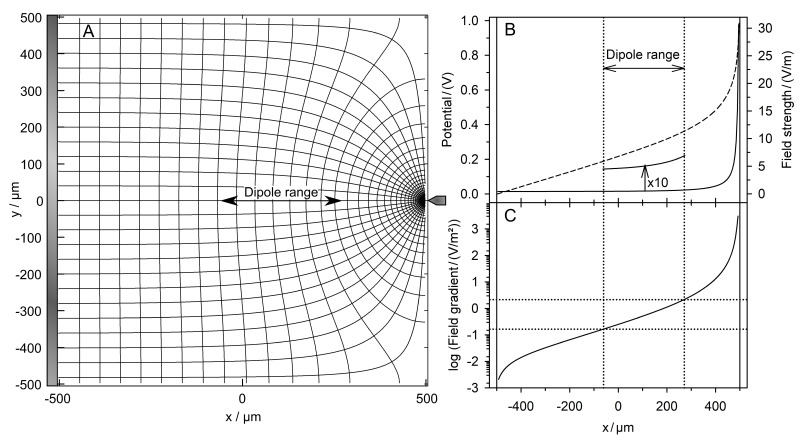
(**A**) Potential and current line distributions in the 1 × 1-mm^2^ chamber without the sphere energized with 1 V at the pointed electrode (center right) versus 0 V at the plain electrode (vertical gray bar on the left). At the symmetry line, the dipole range is marked. (**B**) Potential (dashed) and field strength (full) along the symmetry line of the chamber (500 µm ≤ x ≤ 500 µm, y = 0 µm). Vertical lines mark the limits of the chamber volume. The curve was enlarged by multiplication with a factor of 10 to show the field behavior in the dipole region more clearly. (**C**): Field gradient along the symmetry line.

**Figure 3 micromachines-13-01002-f003:**
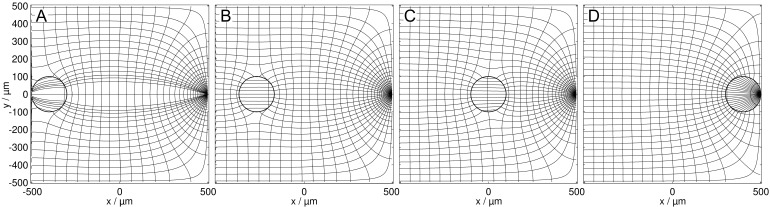
Potential and current line distributions for different positions of the 1.0 S sphere in 0.1 S medium, in front of the plain electrode (**A**), on the watershed (**B**), in a largely homogeneous field region (**C**), and at the pointed electrode (**D**). The conductances are (**A**): 35.744 mS, (**B**): 35.563 mS, (**C**): 35.647 mS, and (**D**): 83.912 mS. The basic sheet conductance $L_{Basic}^{2D}$ of 34.908 mS without a sphere corresponds to a cell constant of $k^{2D}=0.34908$.

**Figure 4 micromachines-13-01002-f004:**
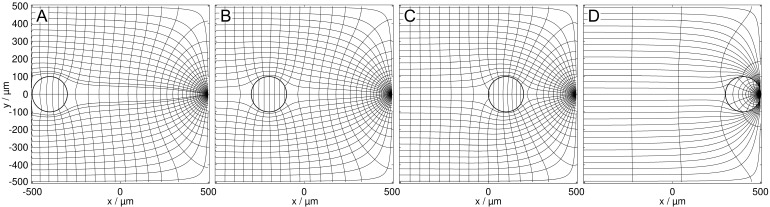
Potential and current line distributions for different positions of the 0.1 S sphere in the 1.0 S medium, at the plain electrode (**A**), in a largely homogeneous field region (**B**,**C**), and in front of the pointed electrode (**D**). The conductances are (**A**): 343.29 mS, (**B**): 342.28 mS, (**C**): 340.10 mS, and (**D**): 60.682 mS. The basic sheet conductance $L_{Basic}^{2D}$ of 348.97 mS without sphere corresponds to a cell constant of $k^{2D}=0.34897$.

**Figure 5 micromachines-13-01002-f005:**
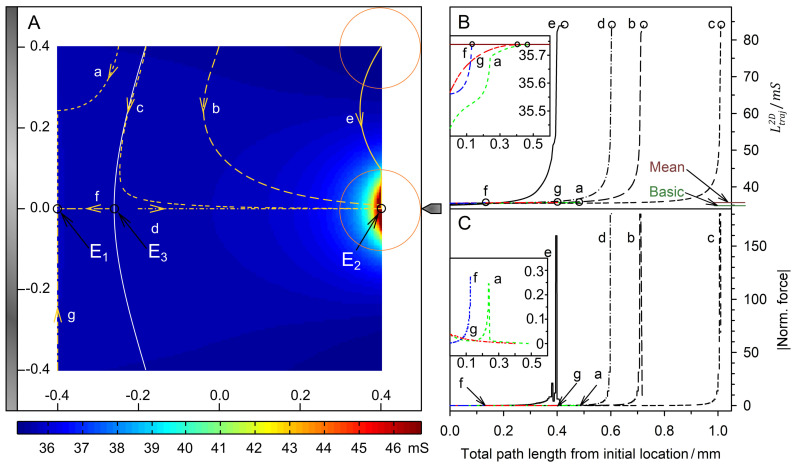
Single 200 µm, 2D sphere of 1.0 S (reddish circles in (**A**)) in the chamber of Figure 2 with 0.1 S medium. The mean conductance is ${\bar{L}}^{2D}=35.739\;\mathrm{mS}$. (**A**) Conductance field plot with trajectories (a–g). A watershed (bent white line) separates the two caption areas of the stable endpoints E_1_ and E_2_. E_3_ is an instable saddle point in the middle of the watershed. (**B**) Sheet conductance along the trajectories. Basic and mean conductance are marked. The system’s sheet conductance increases steadily along each trajectory, reaching moderate and high peak values at the endpoints E_2_ and E_1_, respectively. Trajectories b, c, d, and e end at E_2_. Trajectories a, f, and g end at E_1_, reaching ${\bar{L}}^{2D}$ by coincidence (insert). (**C**) Normalized DEP forces calculated with Equation (16) from the conductance values in (**B**).

**Figure 6 micromachines-13-01002-f006:**
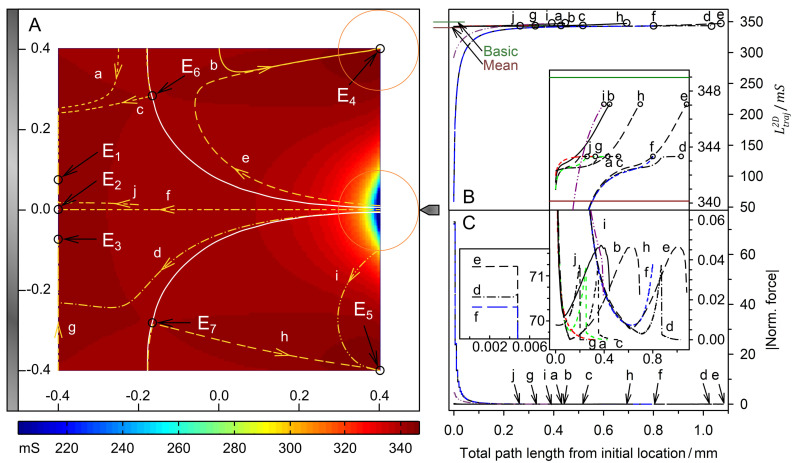
Single 200 µm 2D sphere of 0.1 S (reddish circles in (**A**)) in the chamber of Figure 2 with 1.0 S medium. The mean conductance is ${\bar{L}}^{2D}=340.14\;\mathrm{mS}$. (**A**) Conductance field plot with trajectories (a–j). Two watersheds (bent white lines) and one symmetry line (trajectory f) separate four catchment areas with the four stable endpoints (E_4_, E_1_, E_3_, and E_5_). E_6_, and E_7_ are instable saddle points. E_2_ is an instable minimum at the end of the symmetry line. Trajectories close to f, such as j, are diverted to E_1_ or E_3_. (**B**) Sheet conductance along the trajectories. Basic and mean conductance are marked. Trajectories b and e end at E_4_; trajectories a, c, and j at E_1_; trajectory f at E_2_; trajectories d and g at E_3_; trajectories i and h at E_5_. (**C**) Normalized DEP force calculated with Equation (16). Forces along trajectories parallel to the plain electrode toward the endpoints E_1_, E_2_ and E_3_ are very low. The constant forces observed at the start of trajectories e, d, and f (Figure 6C, left insert) may be due to the reversal of the effect discussed above.

**Figure 7 micromachines-13-01002-f007:**
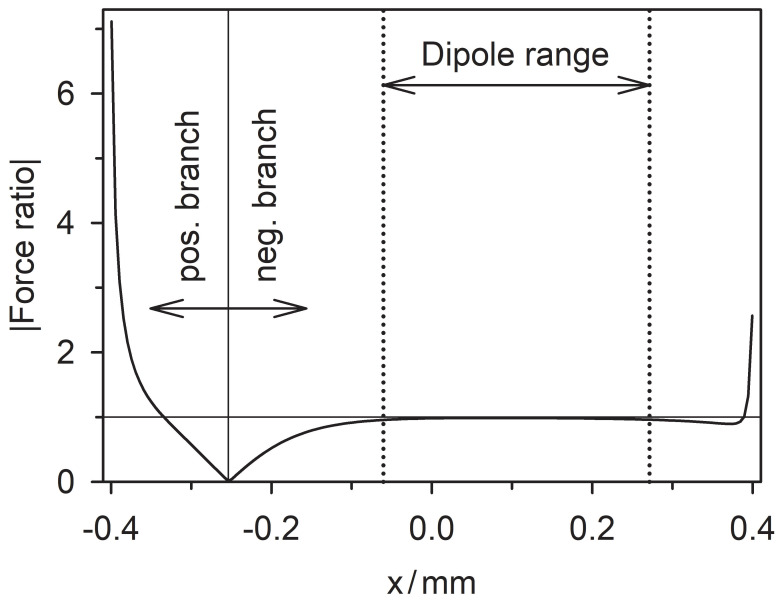
Absolute value of the quotient of the normalized DEP forces acting on the 1.0 S and 0.1 S spheres plotted along the symmetry line of the chambers (trajectories d and f of Figure 5 and trajectory f of Figure 6).

**Figure 8 micromachines-13-01002-f008:**
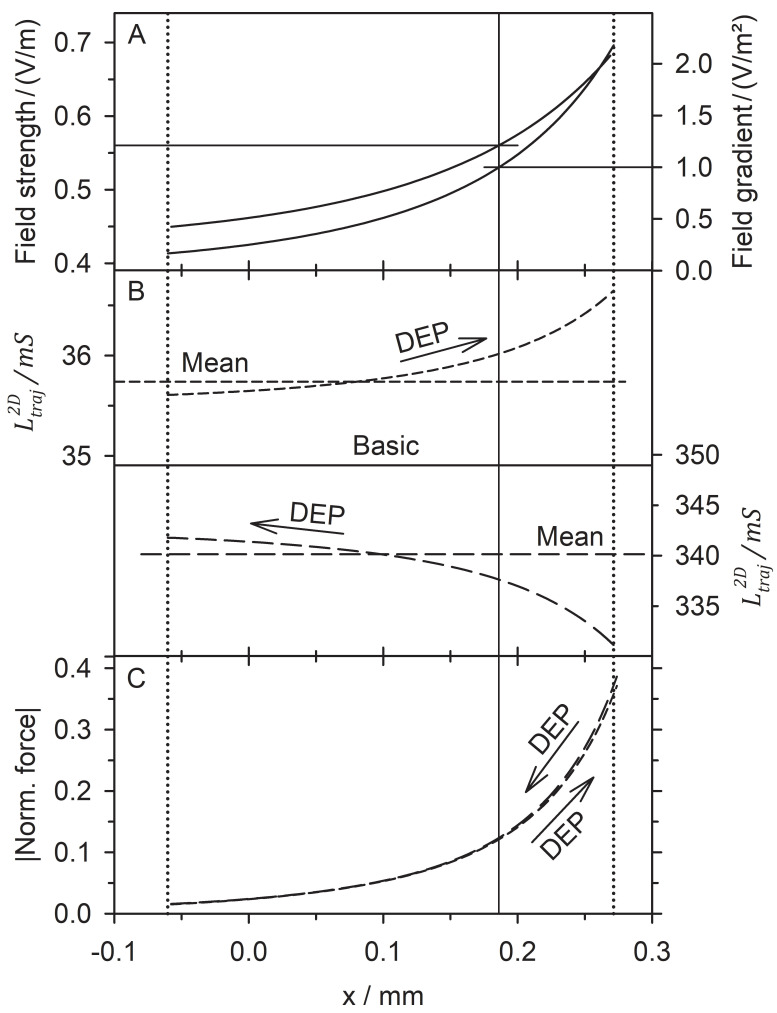
DEP behavior in the dipole range marked in Figure 2. (**A**) Field strength and field gradient along the symmetry line for 1 V potential difference at the electrodes. The vertical auxiliary line at x = 186 µm is perpendicular at a field gradient of 1 V/m^2^ (field strength of 0.5602 V/m). Horizontal auxiliary lines run out to the respective ordinates from the intersections with field strength and field gradient plots. (**B**) DEP of the sphere increases the overall conductance of the chamber. Top: 1.0 S sphere, 0.1 S medium (short, dashed line, left ordinate, positive DEP), bottom: 0.1 S sphere 1.0 S medium (long dashed line, right ordinate, negative DEP). At x = 186 µm, the conductances are 36.02 mS and 337.6 mS. The common baseline corresponds to the basic conductances of the two setups. (**C**) At x = 186 µm, the normalized forces from Equation (16) are 0.1211 (1.0 S sphere, 0.1 S medium) and 0.1234 (0.1 S sphere, 1.0 S medium).

## Data Availability

Not applicable.

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
