# Peer review of "Dielectrophoresis from the System’s Point of View: A Tale of Inhomogeneous Object Polarization, Mirror Charges, High Repelling and Snap-to-Surface Forces and Complex Trajectories Featuring Bifurcation Points and Watersheds"

_micromachines, 2022, doi:10.3390/mi13071002_

Round 1

Reviewer 1 Report

REVIEWER COMMENTS

 Manuscript title: Dielectrophoresis from the system’s point of view: A tale of in-homogeneous object polarization, mirror charges, very high forces and complex trajectories featuring bifurcation points and watersheds

The manuscript presents a theoretical and modeling study on dielectrophoresis. It explores the topic of dielectrophoresis from the fundamentals. The novel aspect of the present article is that, perhaps for the first time, dielectrophoresis is being analyzed from the system’s perspective, not from the object’s perspective. The authors are absolutely correct that in most studies the focus is on calculating the dielectrophoretic force exerted onto an object, usually to predict how an object will behave or migrate under the effects of a dielectrophoretic force. I have not seen any study where the effect of the dielectrophoretic force on the entire system is analyzed. The present study also offers an explanation for the experimental observation of dielectrophoretic trapping or accumulation of small particles (proteins and viruses) in systems where the traditional dipole approach cannot explain how trapping and accumulation can overcome Brownian motion. This is a solid and valuable study that will make a strong contribution to the field of electrokinetics in general.

After reading the manuscript, it is the opinion of this reviewer that this is a well-written and well-organized article. I just found a couple of minor typos that should be corrected (list below, that it is not comprehensive). I found the manuscript quite enlightening. Just to provide one example, section 2.1 discusses the FCM of spherical particles, where the values of FCM were found to range from -1.5 to 3.0 for a 3D sphere, which is different from the commonly used range of -0.5 to 1.0. Another important aspect is that this work considers the size of the object in reference to the size of the system/chamber, which is rarely done, and as stated by the authors, in some cases the objects can be relatively large compared to the chamber size.

My only suggestion would be to discuss more reference 17, which is the recent paper from Professor Pethig regarding the use of FCM.  Can the authors add just a couple of sentences discussing the differences/similarities between their proposed approach and the approach suggested by Professor Pethig?

L=Line number

·         L35, there is an extra “]” after reference 1.

·         Some headings have all wards starting with uppercase letters and some do not – my guess is that the former style should be followed for all headings.

·         List of references: Some journal titles are written all in uppercase letters.

Author Response

Dear Reviewer,

Thank you for your time and the very positive and helpful comments.

We have highlighted the changes made in our manuscript.

In response to your comments and an email exchange with Prof. R. Pethig, who suggested that we clarify the reason for the different magnitudes of our Clausius-Mossotti factor, we have added a brief comment under Eq. 2. As suggested by him, we have refrained from a discussion of his work. We think that publishing it in the same DEP issue of Micromachines ensures the common perception of his and our contribution.

The typos have been corrected.

Sincerely,

Jan Gimsa

Reviewer 2 Report

The manuscript presents two-dimensional calculations of the response of a spherical particles in an inhomogeneous electrical field. It is based on an analytical model recently published by one of the authors.

The matter is interesting and should be published. However, the authors should take into account the following points.

1. line 26-27: "We believe that our approach can explain experimental findings such as the accumulation of viruses and proteins..."

The present model with a size ratio between chamber and object of around 5 (39 voxels / 199 voxels) does not seem to be applicable to protein accumulation with protein diameters of typically 4 nm and thicknesses of photolithographical electrodes of 100 nm.

Please make this clear in the text.

2. line 39: "a steady increase in the overall polarizability..."

It remains unclear on which parameter this increase depends. On time?

3. line 58: " ...is typical, because the objects are relatively large with respect to the chamber [10–15]."

Although the references given support this view, the authors should reconsider whether this can be really generalized to this extent.

line 69: "reactive and active components"

Please explain these crucial terms where they appear in the text first, not as late as, e.g., in lines 158-161.

4. line 108: "Complex parameters are underscored." This writing appears somewhat deviating from the common use of an asterisk superscript and, hence, confusing.

5. line 131-132: The authors' use "L" for conductance is somewhat confusing since the letter commonly used is "G". Is there a special reason for this? If not, common symbols should be used.

6.line 141-142: Better use "RMS" than "eff".

7. line 148-149: What is the physical meaning of delta epsilon DEP?

8. line 190-194: It appears somewhat "magical" that the force in eqn 16 is expressed in "m" and then can be transformed into "Newton".

This is confusing.

9. line 211-212: "a square chamber of x=y=1m confined by plain-parallel electrodes with a depth of 1 m perpendicular to the sheet plane"

How can a "square chamber" have a depth?

This is confusing.

10. line 224: Please give reasons for your choice of a diameter of 39 voxels. This appears rather arbitrary.

11. line 253: " DEP System with a Homogeneous Sphere"

Throughout the manuscript the authors claim to deal with a sphere. However, only a 2D case is dealt with. Accordingly, when they draw conclusions for a 3D case this can be done only for a cylinder with its axis in z-direction. Please make this clear in the text.

12. line 362-363: "The existence of the watershed separating two regions of attraction along the symmetry line contradicts the dipole view."

Is there any experimental evidence for this in the literature?

13. line 367-368: "The peak forces calculated for the 0.1-S sphere are generally lower than for the 1.0-S sphere."

Please discuss this result in the light of conventional DEP theory.

line 420-421: "Moreover, at the pointed electrode, this sphere experiences a force magnitude of around twice as high as the low-conductance sphere."

Please explain in the text in which sense this corresponds to the CM factor showing the same behaviour changing from +1 for positive DEP to -0.5 for negative DEP.

Author Response

Reply to reviewer #2

Dear Reviewer,

Thank you for your time and the helpful comments.

We have highlighted the changes made in our manuscript.

Below are our responses to your points in the order in which you raised them.

Sincerely,

Jan Gimsa

The present model with a size ratio between chamber and object of around 5 (39 voxels / 199 voxels) does not seem to be applicable to protein accumulation with protein diameters of typically 4 nm and thicknesses of photolithographical electrodes of 100 nm.

Please make this clear in the text.

Thank you for making this point. The essence of our new model in terms of the high DEP forces is the inhomogeneous polarization of the objects, which also occurs in the electrical interaction of two or many objects of similar size. Moreover, the force increase also occurs at the plain electrode (Fig. 5, trajectories a and f). It is clear that this force would not be different for larger electrodes, since it results from the inhomogeneous polarization of the object, which would not change even if the object is placed in the center of a much wider smooth electrode (Fig. 3A).

We have checked that all effects are very similar for objects with 19 voxels and even with 3 voxels. With respect to a ratio of 19 voxels for the object and 199 voxels for the simple electrode (1:10.5), your argument (4 nm:100 nm=1:25) is not very strong. However, we found that the approximation of the spherical object size and the resolution of the current and equipotential lines inside the objects is not satisfactory with a smaller number of object voxels. We added the number of voxels for the tested objects with a different size at the end of 3.2.

Please also see our reply to your point 10.

  1. line 39: "a steady increase in the overall polarizability..."

It remains unclear on which parameter this increase depends. On time?

You are right, “steady” refers to time (See also Fig. 6 in ref. [3]).

  1. line 58: " ...is typical, because the objects are relatively large with respect to the chamber [10–15]."

Although the references given support this view, the authors should reconsider whether this can be really generalized to this extent.

It has not become entirely clear to us what the reviewer is referring to. It is clear that whole microfluidic systems are generally much larger than the objects. On the other hand, the volumes used for DEP manipulations are much smaller than the whole system. Therefore, we believe that "typical" is the correct term here.

line 69: "reactive and active components"

Please explain these crucial terms where they appear in the text first, not as late as, e.g., in lines 158-161.

You are right, reactive and active components are crucial terms and of general importance for AC electrical phenomena. Therefore, the terms should be generally understood by those working in the field. We are aware that the concept of reactive and active contributions is new to DEP. However, a clearer picture requires further investigation and cannot be explained in brief. Therefore, we decided to use model parameters that avoid the occurrence of reactive components.

In response to your comment, we have added a brief statement in the introduction: " “Our derivations have shown the importance of distinguishing between active and reactive components in DEP [3].  In the case of suspensions, the reactive components of the impedance result in extraordinarily high permittivities and conductivities at low and high frequencies, respectively.  We suspect that this stealth effect has prevented - for over a century - any discussion of AC-electrokinetic forces in terms of the electrical work performed on suspensions.”

We have chosen DC properties to avoid a lengthy discussion of this problem, which will undoubtedly be necessary in the future. For now, we refer the reader to ref. [3], where, to our knowledge, the problem is discussed for the first time for DEP. We have included this reference in the sentence where the terms are first used.

  1. line 108: "Complex parameters are underscored." This writing appears somewhat deviating from the common use of an asterisk superscript and, hence, confusing.

This notation is used in physics and electrical engineering where the asterisk denotes conjugate complex terms that are also needed for DEP, especially when DEP and ROT expressions are derived in relation to each other. The underscore notation is very helpful to avoid a confusing number of different sub- and superscripts.

  1. line 131-132: The authors' use "L" for conductance is somewhat confusing since the letter commonly used is "G". Is there a special reason for this? If not, common symbols should be used.

The "L" refers to the capital letter lambda, which stands for conductance in physical chemistry, while the "G" stands for Gibbs free energy. As you probably noticed, our manuscript has a "thermodynamic touch".  The use of "G" could lead to confusion in future work.

  1. line 141-142: Better use "RMS" than "eff". ??

“effective” has been changed to “rms”

  1. line 148-149: What is the physical meaning of delta epsilon DEP?

“Delta epsilon DEP” corresponds to “delta epsilon active”. We believe that Eq. 9, along with the discussion of active and reactive contributions to the DEP force, explains the significance.

  1. line 190-194: It appears somewhat "magical" that the force in eqn 16 is expressed in "m" and then can be transformed into "Newton".

This is confusing.

As indicated under the equation, the force is not output in "Newton", but as a normalized force, which is then used in the diagrams. Under 4.7 it is explained how the normalized force can be converted into an actual "Newton" force.

  1. line 211-212: "a square chamber of x=y=1m confined by plain-parallel electrodes with a depth of 1 m perpendicular to the sheet plane"

How can a "square chamber" have a depth?

This is confusing.

You are right, it is as confusing as a round tube having a length. J  Using the same area conductance, in 2D an infinitely thin layer (cf. to a metal layer on glass) would give the same results as the 1 meter deep chamber. However, if we assume a depth of one meter, the relationships between 2D and 3D units become clearer.

  1. line 224: Please give reasons for your choice of a diameter of 39 voxels. This appears rather arbitrary.

The main reason for the relatively low resolution of the overall system was computation time. However, fewer voxels would have resulted in a more ill-shaped "sphere". As explained in 3.2, the odd voxel number of sphere and electrodes defines single central voxel for the sphere and allows precise localization of the sphere with respect to the pointed one-voxel electrode.  Using 19-voxel and 99-voxel spheres, we verified that the results in this size range do not qualitatively depend on the size of the sphere. Even the behavior of a 3-voxel object was qualitatively very similar, even though the object did not really resemble the spherical shape. However, smaller spheres lead to lower resolution, especially for the field inside the object, which would make the effects more difficult to show and explain.

  1. line 253: " DEP System with a Homogeneous Sphere"

Throughout the manuscript the authors claim to deal with a sphere. However, only a 2D case is dealt with. Accordingly, when they draw conclusions for a 3D case this can be done only for a cylinder with its axis in z-direction. Please make this clear in the text.

We have made it clear that we used a 2D sphere. Even if you are right in principle, your conclusion is still not correct. We have seen the same effects in 3D COMSOL simulations using 3D spheres (results not published). We would also like to draw the reviewer's attention to ref. [30], which shows that the results of 3D and 2D spheres are very similar and how they can be converted into each other.

  1. line 362-363: "The existence of the watershed separating two regions of attraction along the symmetry line contradicts the dipole view."

Is there any experimental evidence for this in the literature?

To our knowledge, no. However, in our own experiments we saw an attraction of highly polarizable cells (pos. DEP mode) to bulky electrodes before we introduced glass chip chambers [Gimsa et al. Dielectrophoresis and electrorotation of neurospora slime and murine myeloma cells. Biophys. J. 1991. 60:749-760].

  1. line 367-368: "The peak forces calculated for the 0.1-S sphere are generally lower than for the 1.0-S sphere."

Please discuss this result in the light of conventional DEP theory.

We assume that by "conventional DEP theory" you mean the dipole approach? This point is discussed using the reversibility criterion in 4.6. As said, we explain the difference by the higher efficiency of the "mirror charge induction" by the higher conducting object at the smooth electrode and stronger forces generated by the inhomogeneous polarization at the pointed electrode, i.e. our main new findings are based on these two effects which are not considered in the conventional DEP theory.

line 420-421: "Moreover, at the pointed electrode, this sphere experiences a force magnitude of around twice as high as the low-conductance sphere."

Please explain in the text in which sense this corresponds to the CM factor showing the same behaviour changing from +1 for positive DEP to -0.5 for negative DEP.

The relationship to the CM factor is represented by Eq. 1. Below Eq. 2, we have added a short paragraph explaining why we have "redefined" the CM factor. Please also refer to the new reference [9]. We believe there is an ongoing discussion on this topic.

Reviewer 3 Report

The paper presents an interesting idea of analyzing DEP systems. The relationship between DEP forces on particle trajectories is discussed. Numerical results are presented to justify the claims. I have the following few minor comments:

1.     Is the proposed model completely deterministic? Brownian motion of particles should introduce some randomness. This would be more prominent for submicron particles. Please comment.

2.     There are other works focusing on DEP force and particle trajectories using different approaches. Some of those should be cited. For example: https://www.mdpi.com/2072-666X/12/10/1265

3.     How would the model hold up for a dynamic DEP system where the DEP force is not constant with time? Please comment. This would be the case for travelling wave DEP and moving DEP. The authors may choose to cite a few papers on moving DEP and travelling DEP. A few papers on the topics are:

a.     https://doi.org/10.1021/ac070810u

b.     https://doi.org/10.1063/5.0049126

4.     It would be interesting to see if the particle trajectories correspond to the path of least energy. Would it be possible to plot the energy of the system along the trajectory path? For comparison, energy for a separate arbitrary path (with the same starting and ending point) could be plotted. This would provide an intuitive understanding of the mechanics.

Author Response

Dear Reviewer,

Thank you for your time and interesting comments. However, they come too early for this new topic.

We have highlighted the changes made in our manuscript.

Below are our responses to your points in the order in which you raised them.

Sincerely,

Jan Gimsa

  1. Is the proposed model completely deterministic? Brownian motion of particles should introduce some randomness. This would be more prominent for submicron particles. Please comment.

Yes, the proposed model is completely deterministic. Brownian motion of the particles is not described. However, a discussion of the mechanism for protein and virus trapping has been included in the Conclusions and Outlook section.

  1. There are other works focusing on DEP force and particle trajectories using different approaches. Some of those should be cited. For example: https://www.mdpi.com/2072-666X/12/10/1265

Thank you for this hint. Your paper is now cited in the Conclusion and Outlook section.

  1. How would the model hold up for a dynamic DEP system where the DEP force is not constant with time? Please comment. This would be the case for travelling wave DEP and moving DEP. The authors may choose to cite a few papers on moving DEP and travelling DEP. A few papers on the topics are:
  2. https://doi.org/10.1021/ac070810u
  3. https://doi.org/10.1063/5.0049126

Thank you for the explanation. We know this because we invented traveling-wave dielectrophoresis [Hagedorn et al. Traveling-wave dielectrophoresis of microparticles. ELECTROPHORESIS. 1992 13, 49-54. https://doi.org/10.1002/elps.1150130110]. J However, a discussion of this topic is beyond the scope of this manuscript because TW-DEP uses AC fields that introduce reactive components. As you have probably noticed, we have intentionally avoided this issue.

  1. It would be interesting to see if the particle trajectories correspond to the path of least energy. Would it be possible to plot the energy of the system along the trajectory path? For comparison, energy for a separate arbitrary path (with the same starting and ending point) could be plotted. This would provide an intuitive understanding of the mechanics.

The manuscript is based on the LMEP. As explained in the text, the trajectories of the spheres follow the gradient of the electric work. Since we use the conductance as a parameter for the polarizability of the system, the Rayleigh dissipation or more simply the Joule heating is maximized along the trajectories, which we find very intuitive. Full details can be seen in Figs. 5B and C and Figs. 6B and C or in the conductance matrices in the supplementary material.